# Vero Cells as a Mammalian Cell Substrate for Human Norovirus

**DOI:** 10.3390/v12040439

**Published:** 2020-04-14

**Authors:** Kyle V. Todd, Ralph A. Tripp

**Affiliations:** Department of Infectious Diseases, College of Veterinary Medicine, University of Georgia, Athens, GA 30602, USA

**Keywords:** norovirus, human norovirus, cell substrate, Vero cells

## Abstract

Human norovirus (HuNoV) is a principal cause of acute gastroenteritis worldwide, particularly in developing countries. Its global prevalence is underscored by more serious morbidity and some mortality in the young (<5 years) and the elderly. To date, there are no licensed vaccines or approved therapeutics for HuNoV, mostly because there are limited cell culture systems and small animal models available. Recently described cell culture systems are not ideal substrates for HuNoV vaccine development because they are not clonal or only support a single strain. In this study, we show Vero cell-based replication of two pandemic GII.4 HuNoV strains and one GII.3 strain and confirm exosome-mediated HuNoV infection in Vero cells. Lastly, we show that trypsin addition to virus cultures or disruption of Vero cell host genes can modestly increase HuNoV replication. These data provide support for Vero cells as a cell culture model for HuNoV.

## 1. Introduction

HuNoV is a member of the *Caliciviridae* family [1] and is a non-enveloped, positive-sense, single-stranded RNA virus [2]. HuNoVs have 7.5–7.7 kb genomes that contain three open reading frames (ORFs) [3]. ORF1 codes for the six nonstructural proteins, in order from the N-terminus to C-terminus: p48, nucleoside-triphosphatase (NTPase), p22, VPg, 3C-like protease (3CL^pro^), and RNA-dependent RNA polymerase (RdRp) [4,5]. Subgenomic RNA, containing ORFs 2 and 3, codes for the major and minor structural proteins, VP1 and VP2 [6]. Noroviruses (NoVs) are subdivided into ten genogroups (GI-GX) based upon sequence homology of VP1 [7]. GI, GII, and to a lesser extent, GIV, GVIII, and GIX viruses infect humans. These genogroups are stratified into genotypes: GI (*n* = 9), GII (*n* = 27), GIV (*n* = 2), GVIII (*n* = 1), and GIX (*n* = 1) [7]. The GII.4 HuNoV strains account for ~70% of HuNoV infections [8]. GII.4 HuNoVs have caused pandemics and are now the major circulating strains [9,10,11]. Currently, a recombinant GII.4 Sydney pandemic strain (GII.P16-GII.4 Sydney) causes the majority of infections, making it the most suitable strain for vaccine development [12,13].

HuNoVs are transmitted by the fecal-oral route causing acute, self-limiting infections typified by vomiting and diarrhea [14,15,16,17]. Considerable quantities of viruses are shed in the feces for several weeks, even after symptoms have resolved [18,19,20,21]. The stability of the viral capsid and a low infectious dose facilitate person-to-person transmission. HuNoVs cause ~700 million infections and ~219,000 deaths annually [22,23,24]. HuNoV infections can be debilitating particularly in developing countries where the young (<5 years), the elderly, and the immunocompromised are most susceptible.

Currently, there are no licensed vaccines or approved therapeutics for HuNoV. This is related to the lack of a characterized and reproducible mammalian cell substrate, a lack of a small animal model that emulates infection and disease, and the absence of methods to properly assess vaccine efficacy or protection [25,26,27]. The most progressed HuNoV vaccine candidates are subunit vaccines generated from virus-like particles (VLPs) [28,29,30,31,32]. Although VLP vaccines appear promising, a well-characterized mammalian cell culture substrate is required for the development of inactivated or live-attenuated HuNoV vaccines [33]. Histo-blood group antigens (HBGAs), which are terminal carbohydrates of lipid- or protein-linked glycan chains, are attachment factors for HuNoV [34]. However, it has been shown that HBGA expression does not make a cell permissive for HuNoV infection [35]. CD300ld/CD300lf have been identified as murine NoV receptors and are the only functional receptors known for NoVs [36,37]. Recently, HuNoVs has been propagated in human intestinal enteroids (HIEs) and in a human Burkitt lymphoma B cell (BJAB) cell line [38,39]. These findings are encouraging, but as HIEs are not a stable or clonal cell line, and have a limited lifespan, HIEs are unqualified for vaccine production. Also, the BJAB cell line has been reported to support only a single strain of HuNoV, require HBGA cell culture supplementation, and has reproducibility issues [39,40], making these cells inadequate for vaccine production. In contrast, Vero cells are a continuous mammalian cell line derived from an African green monkey cell line deficient for interferon-α (IFNα) and -β (IFNβ) due to a fortuitous genetic deletion [41,42]. This feature has made Vero cells a leading cell line to use for poliovirus, rabies virus, influenza virus, and rotavirus vaccine propagation [43]. However, past attempts to propagate HuNoVs in Vero cells have been ineffective [38,44,45] possibly because the earlier studies used inadequate virus incubation times. In contrast, this study shows that Vero cells can function as a mammalian cell substrate for HuNoV. Specifically, this study shows that HuNoV modestly replicates in Vero cells as determined by indirect ELISA and quantitative reverse-transcriptase PCR (qRT-PCR) endpoint assays. We examined HuNoV genome replication of two pandemic GII.4 strains and one GII.3 strain by qRT-PCR and using indirect ELISA, flow cytometry, and immunofluorescence show that both structural and nonstructural HuNoV protein levels are increased. Additionally, we show that exosome-mediated HuNoV infection of Vero cells occurs as previously reported for rotaviruses and NoVs [46]. Vero cells were permissive for both filtered and unfiltered clinical stool samples at a wide MOI range. We also explored ways to increase HuNoV replication and show that HuNoV replication can be improved ~1.5-fold by addition of trypsin to the cell culture media, or by Vero cell gene knockdown (KD) or knockout (KO) of specific host genes. These findings provide support for use of Vero cells as a cell culture model for HuNoV replication.

## 2. Materials and Methods

### 2.1. Cells

Vero cells (African green monkey kidney) were obtained from the American Type Culture Collection (ATCC; CCL81.4; lot #738812; Manassas, VA, USA) and cultured at low passage at 37 °C/5% CO_2_ in Dulbecco’s Modified Eagle’s Media (DMEM; GIBCO, Gaithersburg, MD, USA) with or without 5% heat-inactivated fetal bovine serum (FBS; HyClone, Logan, UT, USA). Caco-2 cells (HTB-37; ATCC) were cultured using DMEM supplemented with 20% FBS. The creation of a master cell line bank ensured that cells with low passage numbers were used for all experiments.

### 2.2. Viruses

Stool samples containing GII.3, GII.4 Sydney, or GII.4 Yerseke HuNoVs, or HuNoV-negative samples were obtained from Murdoch Children’s Research Institute (MCRI; Melbourne Victoria, AUS) or the Viral Gastroenteritis Branch in the Division of Viral Diseases (DVD) at the Centers for Disease Control and Prevention (CDC, Atlanta, GA, USA) and stored at −80 °C upon receipt. Stool samples were confirmed to only have GII HuNoV using a TaqMan Array that evaluated 53 enteropathogen species and subspecies [47,48] and sequenced followed by BLAST analysis. The stool samples were thawed on ice before making 10% (*w*/*v*) dilutions in sterile phosphate-buffered saline (PBS; Hyclone). All samples were centrifuged 2× at 1500× *g* for 10 min at 4 °C then 5000× *g* for 10 min at 4 °C. Stool dilutions were passed through 100 µm and 40 µm cell strainers and filtered samples were passed through 0.45 µm or 0.20 µm filters (GE Healthcare, Chicago, IL, USA) before aliquots were made and stored at −80 °C until use.

### 2.3. HuNoV Genome Equivalents (g.e.) Quantification by qRT-PCR

Processed stool samples were treated with RNAzol (Molecular Research Center Inc., Cincinnati, OH, USA) according to the manufacturer’s instructions to generate total RNA to be used for amplification and detection by qRT-PCR. Integrated DNA Technologies (IDT; Coralville, IA, USA) synthesized GII NoV-specific DNA primers and probes to be used with an AgPath-ID One-Step RT-PCR kit (Thermo Fisher Scientific, Waltham, MA, USA). The primers: NK2P_2_F (+) 5′–ATGTTCAGATGGATGAGATTCTC and NKP_2_R (−) 5′–TCGACGCCATCTTCATTCAC were used to amplify a segment of the HuNoV RdRp at a final reaction concentration of [300 nM] [49]. Probe-based amplification was detected by RING_2_-TP 5′–FAM–TGGGAGGGCGATCGCAATCT–BHQ at a final reaction concentration of [120 nM] [49]. 4.25 µL of each RNA sample were used in final reaction volumes of 12.5 µL. Reverse transcription and PCR amplification were carried out using a Mx3005P qPCR System (Agilent, Santa Clara, CA, USA) under the following cycling conditions: 45 °C for 10 min, 95 °C for 10 min, followed by 40 cycles of 95 °C for 15 s, 50 °C for 30 s, and 60 °C for 30 s. The resulting RdRp qRT-PCR levels are considered genome equivalents (g.e.) because the amplified site occurs once in each full-length genome. The g.e. RNA levels for the experimental time-points were divided by the mean g.e. RNA levels from the input time-point (i.e., 0 h) to calculate fold-increases, normalizing the fold-change of input time-points to 1.

### 2.4. ORF1–2 Junction RT-PCR

Total RNA extracted from Vero cells infected with HuNoV at 0 h and 72 h time-points were reverse transcribed and PCR amplified using an AgPath-ID One-Step RT-PCR kit and the NK2P_2_F (+) and Cap C (−) 5′–CCTTYCCAKWTCCCAYGG [1 µM] primers to amplify a segment of the genome between the RdRp and VP1 gene sequences that spans the ORF1–2 junction [50]. PCR amplified products were run on a 1% agarose gel and visualized using ethidium bromide on a FluorChem E system (Bio-Techne, Minneapolis, MN, USA).

### 2.5. qRT-PCR Standard Curve and Controls

A HuNoV dsDNA standard was generated by synthesizing a 100 bp sequence that encompassed the primer and amplification sites (IDT). For each qRT-PCR run, standards containing 10^1^, 10^2^, 10^3^, 10^4^, 10^5^, 10^6^, and 10^7^ copies/well of the amplified sequence were performed in triplicate. For experimental infection qRT-PCRs, 100 ng of Vero cell RNA was added to each standard. Additionally, no template controls (Vero cell RNA only) and no RT/polymerase controls were run in triplicate for each experiment. Wells containing unknown quantities of HuNoV were plotted against the standard curve to determine their g.e. MOI was calculated as the ratio of input g.e. to the number of cells.

### 2.6. HuNoV qRT-PCR

Vero cells were mock-infected or infected (MOI = 1.0) with HuNoV in serum-free DMEM (SF-DMEM) before incubation at 37 °C/5% CO_2_ for either 1 h or 6 h. The virus was removed from all wells except virus input controls and the cells were washed 2× with PBS. RNAzol was used to extract RNA from the cellular fractions as described above. The intracellular levels of HuNoV were determined by dividing the cellular fraction g.e. by the total virus input controls across triplicate experimental and control wells.

### 2.7. H-Antigen ELISA

Vero cells or Caco-2 cells were grown in DMEM + 5% or DMEM + 20% FBS, respectively. Caco-2 cells were differentiated for 21 days before performing the ELISA. The media was decanted and cold 4% paraformaldehyde was added to each well and incubated for 30 min at room temperature (RT). After fixing, the cells were washed 1× with PBS and non-specific binding was blocked by the addition of PBS + 5% BSA for 1 h at RT. Following incubation, a 1:1000 dilution of a mouse IgM anti-blood group H-antigen antibody (Santa Cruz Biotech, Santa Cruz, CA, USA) in PBS + 5% BSA was added to the cells overnight at 4 °C. The solution was decanted and the cells were washed 3× with KPL wash buffer (Seracare, Milford, MA, USA). A 1:2000 dilution of HRP-conjugated goat anti-mouse IgM antibody (Thermo Fisher Scientific) in PBS + 5% BSA was added to the cells for 1 h at RT. The solution was decanted and washed 3× with KPL wash buffer then a 1-step TMB ELISA Substrate Solution (Thermo Fisher Scientific) was used for colorimetric visualization, before determining the optical densities (OD) at 450 nm on an Epoch Microplate Spectrophotometer (Biotek, Winooski, VT, USA).

### 2.8. CD300ld/CD300lf ELISA and Immunofluorescence Assay (IFA)

The H-antigen ELISA protocol was used with the following modifications: (1) a 1:100 dilution of a polyclonal rabbit anti-CD300lf antibody (Lifespan Biosciences, Seattle, WA, USA) was used as the primary antibody, and (2) a 1:1000 dilution of either a goat anti-rabbit IgG antibody conjugated to HRP (Thermo Fisher Scientific) or a goat anti-rabbit IgG (H + L) antibody conjugated to Alexa Fluor 488 (Thermo Fisher Scientific) was used as the secondary antibody. The primary antibody reacts with the ectodomains of both CD300lf and CD300ld, because their amino acid sequences are highly conserved [37], meaning the ELISA and IFA likely detect both proteins.

### 2.9. HuNoV Infection

Cell-free supernatants from stool samples, either filtered or unfiltered, were used for infections. A virus master mixture was generated by normalizing the volume of filtered or unfiltered HuNoV for each infection condition using SF-DMEM. The media from Vero cells was decanted and infections were performed with MOI = 1.0 HuNoV g.e., unless otherwise indicated. The plated cells were gently rocked before incubation at 37 °C/5% CO_2_. At each experimental time-point, the HuNoV g.e. were evaluated as described above. Although included for every experiment, uninfected controls are not graphed to improve the clarity of the graphed data, and because no changes were observed in these samples. For HuNoV infections including trypsin, 0.25% trypsin (Gibo by Life Technologies; Thermo Fisher Scientific) was added to the virus at a final infection concentration of 1% [8.8 BAEE units] for 30 min at 37 °C/5% CO_2_ before infection. For HuNoV infections including bile acids, porcine bile extract (0.015–0.045%) (MilliporeSigma, Burlington, MA, USA), or glycodeoxycholic acid (100 µM–1 mM) (MilliporeSigma) were added at the time of infection. For HuNoV passage experiments, Vero cells were infected at MOI = 1.0 with native stool, freeze-thawed 2× at the indicated time-point, then freeze-thawed supernatants were transferred to fresh Vero cells for 72 h.

### 2.10. HuNoV Inactivation

UV irradiation (Spectronics, Westbury, NY, USA) of HuNoV was performed for 1 h at RT. Controls included foil-covered HuNoV to prevent inactivation, which was placed near the UV lamp for 1 h at RT.

### 2.11. HuNoV Replication ELISA

HuNoV-infected Vero cells were freeze-thawed 2× then the supernatants were transferred to high binding ELISA plates (Corning, Corning, NY, USA) and incubated at 4 °C overnight. The plates were washed 3× with PBS before blocking 2× with SuperBlock (Thermo Fisher Scientific) for 15 min at RT. Following the incubation, a 1:1000 dilution of either a polyclonal rabbit IgG anti-VP1 antibody (Abcam, Cambridge, UK) or a polyclonal rabbit IgG anti-p48 antibody (gift from Christiane Wobus) in SuperBlock was added to the wells for 1 h at 37 °C. After removal of the solution, the wells were washed 3× with KPL wash buffer and a 1:1000 dilution of HRP-conjugated goat anti-rabbit IgG antibody (Thermo Fisher Scientific) in SuperBlock was added to the wells and incubated at 37 °C for 1 h. The solution was decanted and the cells were washed 3× with KPL wash buffer and a TMB ELISA Substrate Solution (Thermo Fisher Scientific) was added for colorimetric visualization. OD_450_ was read using an Epoch Microplate Spectrophotometer (Biotek).

### 2.12. Flow Cytometry and IFA

10^5^ Vero cells were infected or mock-infected in a 24-well plate for 72 h, the media decanted, and the cells washed 1× with PBS before 0.05% trypsin addition for 10 min at 37 °C /5% CO_2_. DMEM + 5% FBS was added to the cells, which were gently pelleted at 200× *g* for 5 min. Supernatants were decanted and the cells suspended in Cytofix (Thermo Fisher Scientific) for 10 min at 4 °C. Cells were again pelleted at 200× *g* for 5 min and washed with PBS + 5% BSA. After washing, the cells were pelleted and resuspended in −20 °C methanol and incubated at 4 °C for 30 min. Following permeabilization, the cells were washed with PBS + 5% BSA and blocked with PBS + 5% BSA for 20 min at 4 °C. Monoclonal mouse IgG anti-VP1 (Abcam) and polyclonal rabbit IgG anti-p48 (gift from Christiane Wobus) antibodies were diluted 1:500 in PBS + 5% BSA and incubated at RT for 30 min. The primary antibodies were removed and the cells were washed 3× then resuspended in a 1:500 dilution of polyclonal PE-conjugated goat anti-mouse IgG (BD Biosciences, Franklin Lakes, NJ, USA) and Alexa Fluor 488-conjugated goat anti-rabbit IgG (Thermo Fisher Scientific) in PBS + 5% BSA for 30 min at RT. The cells were washed 3× in PBS + 5% and resuspended in PBS + 5% BSA and analyzed on a LSR II flow cytometer (Thermo Fisher Scientific). Data analysis was performed using FlowJo with 10,000 cells/condition. For IFA, the flow cytometry protocol was used with the following modifications: (1) cells were not trypsinized, (2) cells were fixed/permeabilized with acetone:methanol (60:40), (3) nuclei were counterstained with 4′,6-diamidino-2-phenylindole (DAPI) (1 µg/mL) (Thermo Fisher Scientific) for 15 min at RT following the secondary antibody incubation, and (4) fluorescence images were acquired at 40× magnification using an EVOS FL imaging system (Thermo Fisher Scientific).

### 2.13. Stool-Derived Exosomes

HuNoV-positive stool samples were treated with Exoquick (System Biosciences, Palo Alto, CA, USA) according to the manufacturer’s instructions. The g.e. content of exosome-associated HuNoV was calculated by qRT-PCR. Exosome samples were tested 10× using a NS300 NanoSight (Spectris, Egham, UK) for the determination of vesicle concentrations and sizes [51].

### 2.14. siRNA Gene Knockdown (KD)

ON-TARGET*plus* siRNAs (Dharmacon, Lafayette, CO, USA) targeting human genes were used to KD gene expression in Vero cells. The siRNAs were previously validated to specifically target and knockdown the host genes in Vero cells [52,53]. Four siRNA duplexes, each targeting distinct gene-specific seed regions, were pooled and transfected simultaneously to enhance gene expression KD. siRNA KD of each host gene was confirmed by qPCR compared to non-targeting control (NTC) siRNA treatments. The siRNAs [50 nM final] were pre-plated in 96 well-plates before incubation with 0.35 µL of Dharmafect 4 (Dharmacon) diluted in Hank’s Balanced Salt Solution (HBSS; Hyclone) for 30 min at RT. Following the incubation, 8 ×10^3^ Vero cells were reverse-transfected in SF-DMEM and incubated at 37 °C/5% CO_2_ overnight. At 16 h post-transfection, the transfection media was replaced with DMEM + 5% FBS. At 48 h post-transfection, the wells were infected with HuNoV for 72 h.

### 2.15. CRISPR-Cas9 Gene Editing

Single-plasmid CRISPR gene editing (MilliporeSigma) was performed in Vero cells. Then, 9 × 10^5^ Vero cells were plated in 6-well plates in DMEM + 5% FBS at 37 °C/5% CO_2_. 12.5 µL of Lipofectamine LTX (Thermo Fisher Scientific) and 3.75 µg of CRISPR DNA were each diluted to a final volume of 100 µL in OPTI-MEM (Thermo Fisher Scientific). After mixing, the Lipofectamine and CRISPR DNA were combined and incubated at RT for 30 min. Meanwhile, the cells were washed 1× with HBSS before the addition of DMEM + 5% FBS. After adding 200 µL of the Lipofectamine and CRISPR DNA mix, the plate was incubated at 37 °C /5% CO_2_ for 6 h before the addition of another 1 mL of DMEM + 5% FBS. After 24 h post-transfection, the media was changed to DMEM + 10% FBS, 100 units/mL of penicillin, 100 µg/mL of streptomycin, and 250 ng/mL of amphotericin B (Thermo Fisher Scientific). Single-cell colonies were sorted using a MoFlo Astrios EQ (Beckman Coulter, Brea, CA, USA) based on GFP expression 48 h post-transfection into 96-well plates. Expansion of the colonies from a 96-well plate format to a 6-well plate format was performed in parallel to allow for genotypic (qRT-PCR followed by Sanger sequencing and next-generation sequencing) and phenotypic (HuNoV infection) validation. Gene KO was confirmed by bp deletion resulting in disruption of the ORF. Additionally, clones containing genes with insertions were excluded from the analysis.

### 2.16. Statistical Analyses

Unpaired two-tailed t-tests and one-way ANOVA with Dunnett’s post hoc tests were performed with 95% confidence intervals using GraphPad Prism. *p*-values < 0.05 were considered significant: * *p* < 0.05, ** *p* < 0.01, *** *p* < 0.001, and **** *p* < 0.0001. Unless otherwise indicated, *n* = 3 wells/condition/experiment from *n* ≥ 3 independent experiments were performed. Error bars represent + standard error of the mean (SEM).

## 3. Results

While others were unsuccessful in propagating HuNoV in Vero cells [38,44,45], we show that low-passed Vero cells can be used as a cell substrate for HuNoV. We previously used low-passaged Vero cells in the development of vaccine substrates for several viruses [52,54]. To begin to evaluate HuNoV infection, we showed that HuNoV RNAs are present within Vero cells following a 1 h or 6 h incubation by qRT-PCR (Figure 1A). HuNoVs use HBGAs, such as the H-antigen, as attachment factors [55,56,57]; however, they are not a requirement for HuNoV infection [58,59,60,61,62]. We examined the level of H-antigen expression on Vero cells or Caco-2 cells and showed that Vero cells do not express the H-antigen by ELISA (Figure 1B). Also, we showed that Vero cells express the murine NoV receptor, CD300ld/CD300lf [36,37,63] (Figure 1C,D). We attempted to inhibit HuNoV entry into Vero cells using antibody blockade of CD300ld/CD300lf, and by saturating potential CD300ld/CD300lf binding sites using a CD300ld/CD300lf ectodomain peptide, but both treatments were unsuccessful [51]. These findings are consistent with the report that CD300ld/CD300lf are not receptors for HuNoV [64].

Following the investigation of HuNoV binding, replication in Vero cells was determined and quantified by examining HuNoV genomes from infected Vero cells by qRT-PCR and RT-PCR (Figure 2 and Appendix A). In these studies, GII.4 Sydney genome replication peaked between 48–72 hpi (Figure 2A) and was ablated by UV irradiation as expected (Figure 2B). The fold-decrease following UV-inactivated HuNoV incubation for 72 h likely corresponds to the degradation of non-replicative HuNoV RNA by Vero cells (Figure 2B). To determine if the findings for GII.4 Sydney were strain-specific, Vero cells were examined for their ability to propagate GII.3 and GII.4 Yerseke HuNoV strains (Figure 2C,D). Of note, replication for these three HuNoV strains produced no detectable cytopathic effects at any MOI or time-points tested. To support replication, HuNoV protein expression was examined for GII.4 Sydney (Figure 3A) and GII.3 (Figure 3B) HuNoV using a rabbit anti-VP1 antibody indirect ELISA. Statistically significant (*p* < 0.0001) dose-dependent detection of VP1 was observed at an input time-point (Figure 3A). Furthermore, the ELISA showed increases in HuNoV capsid and p48, a nonstructural protein (Figure 3A,B, and Appendix A). ELISAs were evaluated instead of Western blots because they are quantitative and allow for the detection of low protein levels. We also examined the expression of VP1, and p48 in HuNoV infected Vero cells by flow cytometry (Figure 3C) and showed both proteins were expressed in a low percentage of cells, a result similar to the findings observed by immunofluorescence assay (Figure 3D).

HuNoV was understood to infect susceptible cells as uncoated virions until a recent report showed that murine NoV- and HuNoV-containing exosomes are central to virus infection and replication [46]. Therefore, we examined exosome isolation from HuNoV-infected stool samples and confirmed that the exosome fractions contained more than one-half (~55%) of all HuNoV g.e. (Appendix A). Our findings indicated that exosome-associated HuNoVs are internalized by Vero cells at levels similar to HuNoV from unseparated stool samples [65] and that exosome-associated GII.4 HuNoVs were replication-competent in Vero cells (Figure 4). This study also examined the tempo and peak of HuNoV replication from filtered or unfiltered stool samples. There were no substantial differences detected suggesting that bacteria or other filterable agents from stool samples did not facilitate HuNoV replication [51], contrary to culturing HuNoVs in BJABs [39]. Additionally, HuNoV replication in Vero cells was similar for a wide MOI range (1.0–100) (Appendix A).

Enteric virus capsids that are primarily transmitted by the fecal-oral route are remarkably durable as they have a role in protecting the viral RNA, particularly when the virus is outside of the host cell. However, these capsids typically require proteolytic cleavage for virus activation, binding, infection, and replication [66,67,68,69,70,71]. Proteolytic cleavage has been examined for HuNoV but trypsin was shown to have little effect [44,72,73] likely because the intact virion is resistant [74]. However, smaller conformations of HuNoV capsid proteins are trypsin-cleavable [75]. Thus, we examined a range (1.76–10,000 BAEE units) of trypsin treatments and determined that HuNoV titers slightly increased (~1.5-fold) using a higher MOI = 100 when co-incubated with 8.8 BAEE units of trypsin throughout the infection time-course (Appendix A). This effect was consistent with the abundance of trypsin-cleavable capsid conformations. Trypsin addition also increased exosome-associated HuNoV replication (~1.5-fold) in a MOI-dependent manner, but at a lower MOI = 1.0 (Appendix A). Selective loading of trypsin-cleaved capsid conformations into exosomes may not occur, explaining the inconsistency of HuNoV replication enhancement at MOI = 100.

To elucidate some of the antiviral Vero cell genes that modify HuNoV replication, we evaluated siRNA knockdown (KD) of host genes previously identified to be important for influenza virus [76], poliovirus [54], and rotavirus [53] replication. KD of several genes in Vero cells, specifically empty spiracles homeobox 2 (*EMX2*), fibroblast growth factor 2 (*FGF2*), neuraminidase 2 (*NEU2*), pyrroline-5-carboxylate reductase 1 (*PYCR1*), RAD51 associated protein 1 (*RAD51AP1*), and Sec61 translocon gamma subunit (*SEC61G*) enhanced HuNoV replication in Vero cells (Figure 5 and Appendix A). When several canonical innate antiviral genes were examined following siRNA KD, specifically interferon regulatory factor 3 (*IRF3*), *IRF7*, interferon induced with helicase C domain 1 (*IFIH1*; MDA5), DExD/H-box helicase 58 (*DDX58*; RIG-I), toll-like receptor 2 (*TLR2*), *TLR3*, and *TLR7,* the findings showed that these genes affect HuNoV replication (Figure 5 and Appendix A).

Based on earlier virus-host gene studies for influenza virus [76], poliovirus [54], and rotavirus [53] replication, we generated several CRISPR-Cas9 gene-edited Vero cell KO lines and focused our studies on Vero cells lacking the leucine-rich repeats and guanylate kinase domain-containing (*LRGUK*) gene for its ability to be a substrate for enhanced GII.3 HuNoV replication (Appendix A). The *LRGUK* gene was screened and validated as an antiviral gene regulating rotavirus replication [53], and interestingly only the *LRGUK* KO Vero cell line, but not *LRGUK* KD improved HuNoV replication (Appendix A). This is likely due to the heterozygous nature of the gene KD using siRNAs or bias of the single cell-derived population for enhanced ability to replicate GII.3 HuNoVs. Collectively, these results demonstrate HuNoV replication of three HuNoV strains in Vero cells. These data provide evidence of HuNoV attachment, internalization, genome replication, and viral protein production. Further, modest replication in Vero cells may be enhanced by trypsin or by siRNA KD or CRISPR-Cas9 gene editing of antiviral host genes. These studies provide support for further investigation of Vero cells as a cell culture model for HuNoV research.

## 4. Discussion

We re-examined HuNoV replication in Vero cells to clarify whether Vero cells could be used in HuNoV vaccine development. We found that for all HuNoV strains tested, virus attachment to Vero cells was low ranging from two to four percent, a finding consistent with a previous report [73]. Cell-surface expression of CD300ld/CD300lf on Vero cells was unexpected because it had only been previously observed on immune cells [37,77], and more recently on tuft cells, an uncommon gastrointestinal epithelial cell type [78]. We attempted to inhibit HuNoV entry into Vero cells using antibody blockade, and by saturation of potential CD300ld/CD300lf binding sites using a CD300ld/CD300lf ectodomain peptide, but both attempts were unsuccessful [51]. We showed that the tempo of HuNoV replication in Vero cells was similar to HIE cells [38], BJAB cells [39], gnotobiotic pigs [79], and gnotobiotic calves [80]. We demonstrated HuNoV replication (≥1.5-fold) in three Vero cell line clones over a 72h time-course [51]. Efforts to detect HuNoV replication in Caco-2 cells, HEp-2 cells, HepG2 cells, INT 407 cells, RAW 264.7 cells, and RAW 264.7+ Caco-2 co-cultured cells using comparable methodologies were unsuccessful (Appendix A). We found HuNoV replication in Vero cells to be reduced when compared to BJAB cells (up to 25-fold), or HIE cells (up to 1000-fold) possibly because HuNoV replication in Vero cells may be limited to a single replication cycle. Interestingly, prolonged incubation of HuNoV with Vero cells impedes their ability to replicate after passaging with no replication observed after 24-h incubation with Vero cells (Appendix A). This may indicate that following virus uncoating, a new infection cycle is not initiated but this has not been experimentally confirmed. Notably, a single HuNoV replication cycle has also been reported for other HuNoV replicon models [27]. A comparable attempt at infecting Vero cells did not observe g.e. increases likely because later time-points (days 4–9 pi) were evaluated [45], which likely missed the peak HuNoV replication (Figure 2A).

HBGA-expressing bacteria belonging to *Escherichia coli*, *Enterobacter cloacae*, *Enterobacter aerogenes, Clostridium difficile*, and others have been implicated in improving HuNoV infection [39,81,82,83]; however, in our studies, filtration of stool samples did not markedly increase HuNoV replication in Vero cells, which were similar to findings for HuNoV replication in HIE cells [38]. These data imply that, unlike BJAB cells, Vero cells do not appear to require exogenous HBGAs to aid HuNoV replication. While trypsin is needed for proteolytic cleavage events affecting some enteric viruses [66,67,68,69,70,71], the HuNoV capsid is understood to be resistant to trypsin-mediated cleavage as trypsin does not impact binding or internalization [72,73,84]. HuNoV capsids contain approximately 15 trypsin cleavage sites; however, each site is inaccessible to the enzyme while in its infectious form [74]. Of note, a trypsin-cleavable GII.3 norovirus virion has been described [85,86]. Smaller conformations of the HuNoV capsid form under alkaline conditions [84,87], such as those in the gastrointestinal tract, and upon exposure to trypsin can be cleaved [75]. Following cleavage, these capsid conformations may have a role in the viral replication cycle [75]. The effect of trypsin on HuNoV titers following infection with native stool may correspond to a high MOI containing increased numbers of immunomodulatory trypsin-cleaved capsid conformations. Trypsin enhancement of exosome-associated HuNoV may occur at a low MOI because of virus clustering within exosomes, but this remains to be explored.

Cell culture media supplemented with bile acids have been shown to aid porcine enteric calicivirus [88,89,90], murine NoV [91], and HuNoV [38,92,93]. Bile acids are thought to assist the virus to escape from endosomes into the cytosol [90,93,94]. Also, bile acids have been shown to bind to murine NoV [91] and HuNoVs in a genotype- or strain-specific manner [95]. It is known that bile acids do not bind all HuNoV strains [95]. Our attempts at using bile or bile acids to enhance GII.4 Sydney replication in Vero cells were unsuccessful (Appendix A).

The antiviral host genes that affect HuNoV replication are not well-understood, but it is known that viruses co-opt host genes to replicate, and host genes are known to be required for virus replication. We took advantage of previous findings of virus–host gene interactions [52,96,97]. Based on genome-wide siRNA screens for several RNA viruses, we examined the top six genes for which KD enhanced virus permissiveness and replication in Vero cells [52]. We created CRISPR-Cas9 gene-edited Vero cell lines to evaluate HuNoV replication based on siRNA KD findings. The antiviral host genes identified have diverse functions. It was shown that EMX2 functions in the Wnt/β-catenin pathway and may have a role in cell cycle regulation and apoptosis [98]. EMX2 has also been shown to bind to the translational factor eIF4E [99]. This interaction may affect HuNoV VPg recruitment of eIF4E, modulating HuNoV replication. FGF2 has been shown to stabilize basally expressed RIG-I [100], implicating its downregulation with decreased virus sensing mechanisms. NEU2 functions as a sialidase and cleaves Lewis X carbohydrates [101]. HuNoV VLPs has been shown to bind Lewis X [57,102,103], conceivably implicating a decrease in *NEU2* gene expression with increases in attachment factor availability. PYCR1 leads to cell cycle arrest [104,105]. A similar mechanism has been shown for the murine NoV VPg [106]. It remains unknown if VPg directs cell cycle arrest through PYCR1 downregulation. RAD51AP1 facilitates homologous recombination of host DNA [107], while SEC61G functions in membrane protein translocation and incorporation into the endoplasmic reticulum [108]. The interactions between HuNoV and these genes are speculative; however IRF3 [109,110], IRF7 [110,111], MDA5 [112,113,114,115], RIG-I [35,111,113,114], TLR3 [112], and TLR7 [114,116,117] have been previously implicated in NoV responses. Of note, TLR2 enhances cytokine production upon the detection of rotavirus [118]. No additive or synergistic effects were observed for any of the genes when two genes were KD simultaneously. We also attempted to corroborate the host genes needed for HuNoV replication with microRNA (miR) regulation, specifically miR-let7 (*DDX58*; RIG-I), miR-18a (*RAD51AP1*), miR-26b (*TLR3*), miR-105 (*TLR2*), miR-223 (*TLR3*), and miR-512 (*TLR7*), but were unsuccessful in linking the miRs to increased HuNoV replication (Appendix A). It is likely that the promiscuous nature of miRs for different targets may have obscured any effects. In summary, infection of Vero cells can occur and these data provide preliminary support for use of Vero cells as a cell culture model. Although replication of HuNoVs in Vero cells is low, infection rates and viral replication are poised to increase dramatically with the impending discovery of the HuNoV receptor. Ectopic expression of the HuNoV receptor in a Vero cell line would provide greatly enhanced HuNoV replication in this model, providing a robust system that is more applicable for industrial HuNoV vaccine production.

## Figures and Tables

**Figure 1 viruses-12-00439-f001:**
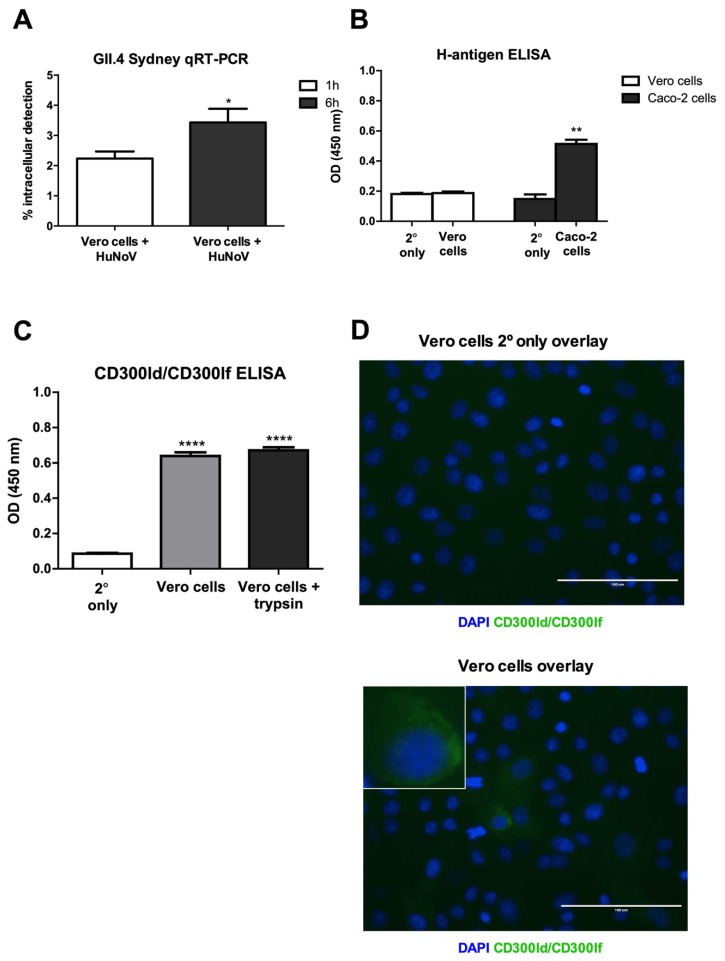
(**A**) GII.4 HuNoV detection by qRT-PCR. qRT-PCR analysis of RNA extracted from HuNoV-infected Vero cells. Data represent *n* = 3 + SEM. * *p* < 0.05. (**B**) Vero cells do not express detectable H-antigen. An indirect ELISA was used to detect the H-antigen on the cell surface of Vero cells and Caco-2 cells. No signal was observed for secondary only antibody controls (2°). Data represent *n* = 4 (Vero cells H-antigen ELISA) or *n* = 1 (Caco-2 cell H-antigen ELISA) + SEM. ** *p* < 0.01. (**C**) Vero cells express CD300ld/CD300lf. Anti-CD300lf (ectodomain) indirect ELISA of Vero cells following 24-h treatment ± 8.8 BAEE of trypsin. No signal was observed for secondary only antibody controls (2°). Data represent *n* = 3 + SEM. **** *p* < 0.0001. (**D**) CD300ld/CD300lf expression on Vero cells by immunofluorescence. A representative image is shown at 40× magnification from *n* = 3 biological replicates. CD300ld/CD300lf-associated fluorescence was not observed for secondary antibody only (2°) treated cells. The inset shows a CD300ld/CD300lf-expressing Vero cell.

**Figure 2 viruses-12-00439-f002:**
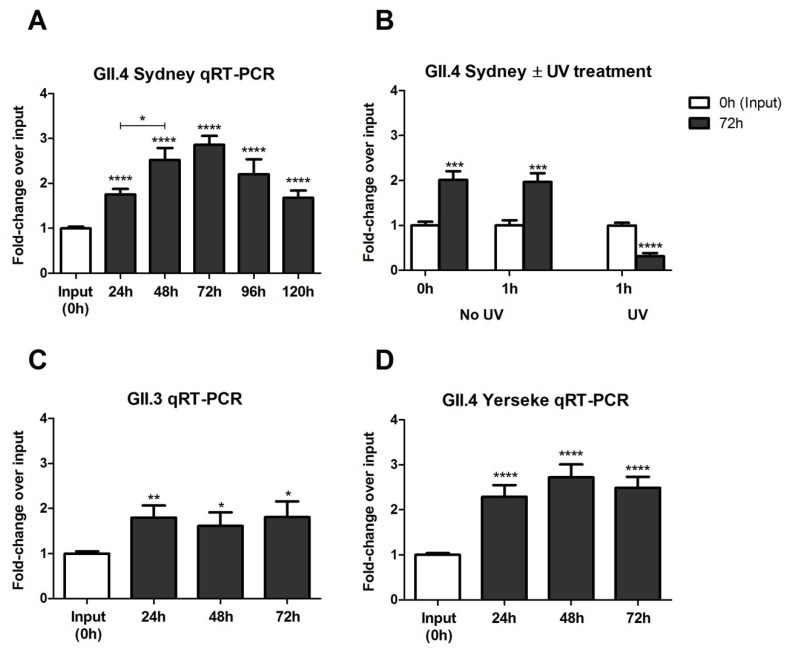
(**A**) Vero cells are permissive for GII.4 Sydney HuNoV. HuNoV levels were examined over a 5-day time-course and peaked between 48–72 hpi by qRT-PCR. Data represent *n* = 3 + SEM. * *p* < 0.05 and **** *p* < 0.0001. (**B**) UV inhibits HuNoV replication. 1 h treatment with UV light inactivated HuNoV reducing replication at 72 hpi as measured by qRT-PCR. Data represent *n* = 3 + SEM. *** *p* < 0.001 and **** *p* < 0.0001. Vero cells are permissive for GII.3 (**C**) and GII.4 Yerseke (**D**) HuNoVs. qRT-PCR analysis over a 3-day time-course. Data represent *n* = 3 + SEM. * *p* < 0.05, ** *p* < 0.01, and **** *p* < 0.0001.

**Figure 3 viruses-12-00439-f003:**
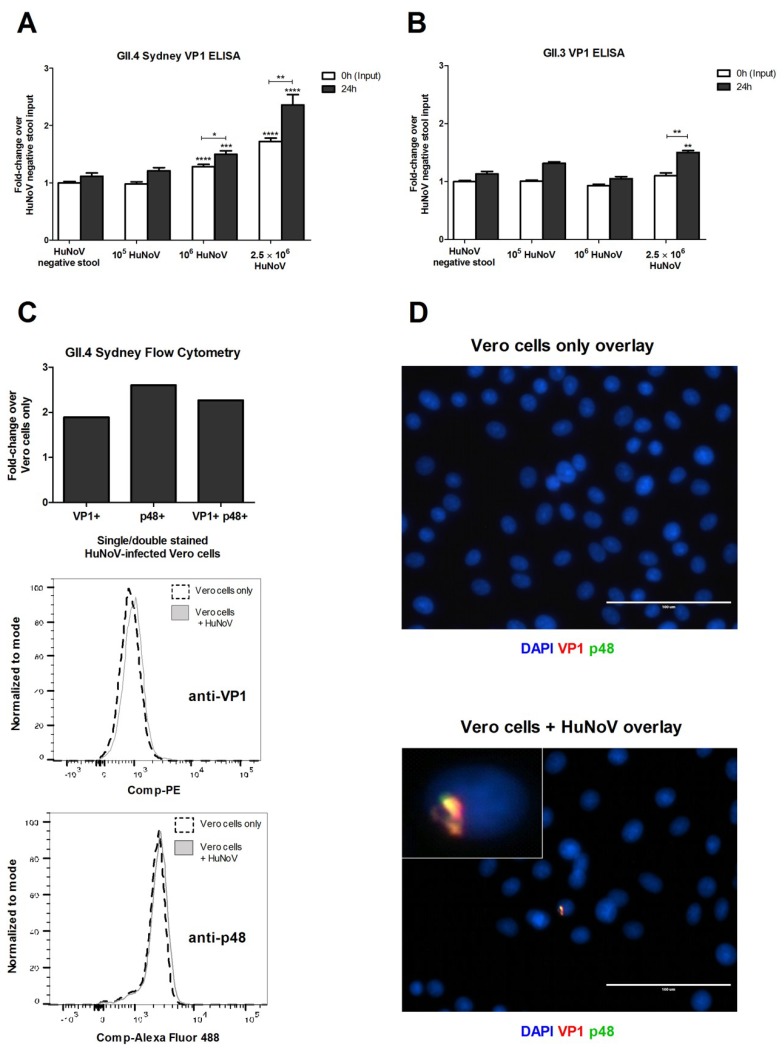
Detection of GII.4 Sydney (**A**) and GII.3 (**B**) HuNoV VP1 by ELISA. OD (450 nm) were measured and fold-changes normalized to HuNoV negative stool input values were graphed. Data represent *n* = 3 (A) or *n* = 1 (B) + SEM. * *p* < 0.05, ** *p* < 0.01, *** *p* < 0.001, and **** *p* < 0.0001. (**C**) Increased VP1 and p48 staining frequency in HuNoV-infected Vero cells. Detection of single-positive (VP1+ or p48+) and double-positive (VP1+ p48+) Vero cell populations following HuNoV inoculation. Histograms generated following fluorescent bead compensation with the cell counts normalized to the mode. Data represent *n* = 1 from 5 biological replicates. (**D**) Visualization of GII.4 Sydney VP1 and p48 proteins by immunofluorescence assay. Representative image is shown at 40× magnification from *n* = 3 biological replicates. VP1- or p48-associated fluorescence were not observed for secondary antibody only (2°) treated cells. The inset shows a HuNoV-infected Vero cell.

**Figure 4 viruses-12-00439-f004:**
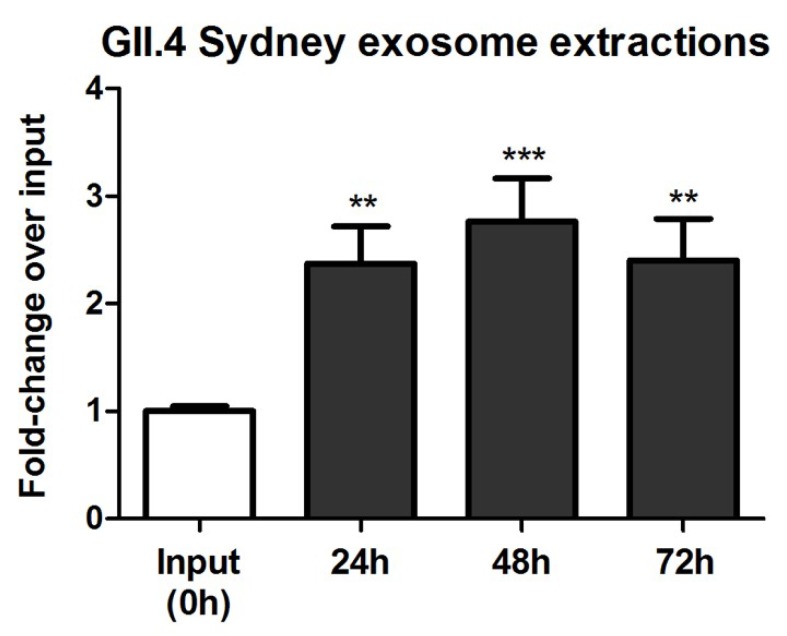
Vero cells are permissive for exosome-associated GII.4 Sydney HuNoV. qRT-PCR analysis of a 3-day time-course. Data represent *n* = 3 + SEM. ** *p* < 0.01 and *** *p* < 0.001.

**Figure 5 viruses-12-00439-f005:**
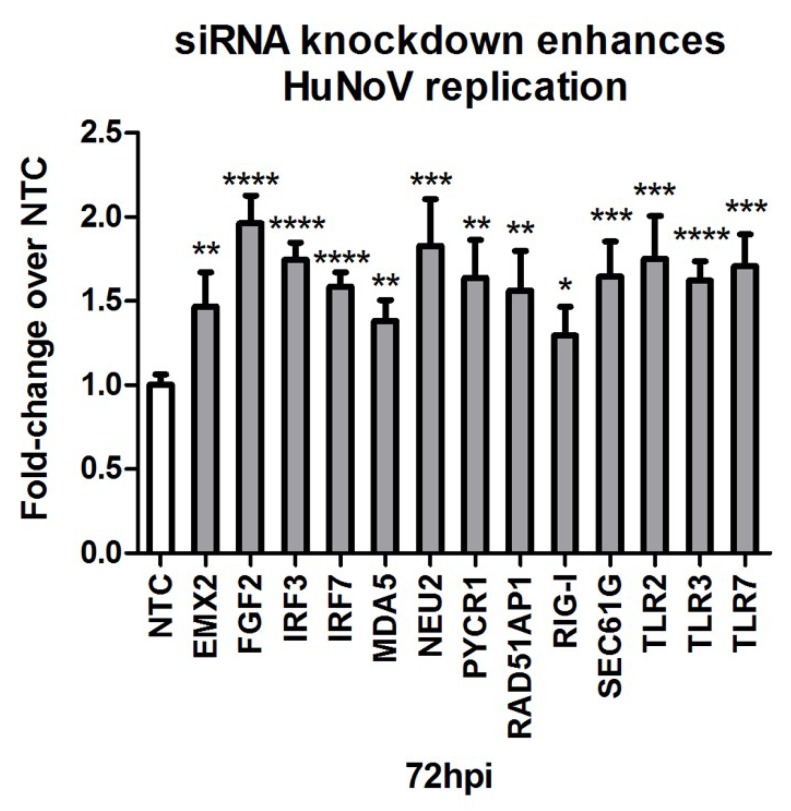
The knockdown of antiviral host gene expression increases GII.4 Sydney HuNoV replication. Treatment with pooled gene-specific siRNAs [50 nM] 48 h before infection resulted in substantial increases in HuNoV titers 72 hpi by qRT-PCR compared to non-targeting control (NTC) siRNA treated cells. Data represent *n* = 3 + SEM. * *p* < 0.05, ** *p* < 0.01, *** *p* < 0.001, and **** *p* < 0.0001.

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
