# Peer review of "Vero Cells as a Mammalian Cell Substrate for Human Norovirus"

_viruses, 2020, doi:10.3390/v12040439_

Round 1

Reviewer 1 Report

Todd et al have investigated the use of Vero cells, a widely used cell line in the field of virology, for the replication of norovirus. They have tested several norovirus strains and explored several methods to increase replication efficiency in this cell line. Although the observed replication is very minimal I do feel that this study is of interest as currently only organoid cultures and B-cell cell lines support norovirus replication. In addition, the authors have elaborately studied the effect adding bacteria, trypsin, the role of endosomes and the knockdown several host genes on norovirus replication. Therefore, this manuscript also provides valuable information that can help research towards a robust norovirus culture model forward. It should however clear to eaders that the observed replication is much lower than for organoids or BJABs

Minor comments

It would be good to discuss the work that Goodfellow’s lab did to increase replication in the organoid model in light of the vero culture model

Lines 16.  The BJAB culture model was tested in multiple laboratory and worked in some but not all. It has not been shown that the vero cell culture model is more robust.

Line 61. I think influenza vaccines are only produced in eggs or on MDCK cells

Line 171 Please provide the specifics for the trypsin that was used (TPCK treated, brand)

Figure 1A Not sure what the authors mean with % intracellular detection, how was that calculated?

Figure 1B/C please describe “R0 only” in the legend

Line 322 This finding is not “contrary to a previous report” since they used a completely different cell line. Please adapt.

Figure 3C The difference between overlays in the histograms is very difficult to see, perhaps a scatterplot would be more clear.

Figure 3C/D are all figures made 24 hours pi?

Line 331 infection should be inoculation

Line 351 But there is a decrease of exosome associated replication at MOI 100 while this should also contain more trypsin-cleavable capsid conformations.

Figure 5 please provide the abbreviation of NTC in the legend.

Figure 5 it is striking that all siRNAs have some positive effect on replication. Did the authors test siRNAs that were unlikely to have an effect to check that the  increase in replication is host gene specific?

Line 440 This is quite a bold statement. Binding to Lewis X (secretor negative) is strain- and genotype-dependent and is it sure that this sialidase does not cut sialic acids from other (secretor positive) Lewis glycans?

Reviewer 2 Report

Except for the murine norovirus (NoV), a reproducible system culture allowing the routinely replication in vitro of NoVs is not still available. To date, there are two published systems (B cell line and stem cell-derived enteroids) supporting replication of human NoVs, but the levels of replication are not sufficient for the generation of highly purified virus stocks or the development of culture-based quantification assays.

The paper that I reviewed " Vero Cells as a Mammalian Cell Substrate for Human Norovirus” by Todd et al. describes the employ of Vero line cell as possible substrate for replication in vitro of HuNoV. The experimental design was performed by using three human strains, NoV GII.3, GII.4 Sydney and GII.4 Yerseke.

In the following study, the Authors demonstrated by quantitative RT-PCR that HuNoV RNA was present within Vero cells following 1h-6h post-incubation. Also, they showed that Vero cells do not express the H-antigen, as well as the murine NoV receptors CD300ld/CD300lf are not involved in the binding of HuNoV. However, in spite of the several molecular and serological evaluations performed by the Authors, reading this manuscript I did not find convincing data demonstrating clearly that the Vero cells may support the replication of NoV in vitro. Whether the NoV replication in Vero cells did not produce CPE, electron microscopic visualization could be used to confirm a productive infection and to visualize virus particles with typical morphology.

Furthermore, the Authors evaluate the presence of the VP1 capsid protein and that of the P48 by IFA, flow cytometry and indirect ELISA. The results obtained by IFA (figure 3D) revealed a very low expression of both proteins, similarly when the infected cells were assessed by flow cytometry. By converse the ELISA seems indicate an increase of proteins expression. The Authors did not carry out Western blotting because less sensitive than ELISA. By converse, I believe that this test could be useful in this step considering its specificity, so confirming the ELISA results.

Round 2

Reviewer 2 Report

Accepted in present form

This manuscript is a resubmission of an earlier submission. The following is a list of the peer review reports and author responses from that submission.

Round 1

Reviewer 1 Report

The authors are claiming that there is replication of human norovirus (NoV) in Vero cells and « these data provide the foundational support for use of Vero cells as a cell culture model » (line 430). I believe readers would clearly disagree with that.

The introduction is well documented and explains that previous assays to grow NoV on Vero cell were unsuccessful. They also discussed the past characterization of NoV ligands, namely the HBGA and the CD300 receptor. Of note, CD300 is probably irrelevant for human NoV since it has been characterized as an MNV receptor, which targets immune cells and not enteric cells.

In this study, several techniques were used to characterize NoV replication. Unfortunately the analysis only focused on Vero cells without providing a point of comparison with another cell type. It is now widely accepted that HBGA is the natural ligand for human NoV. It is strange that monkey-derived cells like the Vero cell line not expressing HBGA are more permissive for NoV growth than Caco-2 cells, which are of human origin and expressing HBGA. It would have been pertinent to run the same experiments on Caco-2 cells for the sake of comparison with the Vero cells. In addition, Table S1 is really unclear.

The major concern about the study is the poor performance of the Vero cells for NoV replication, while at the same time the authors argued that enteroids showed limitations for NoV growth. That being said, we have no clear demonstration about the replication into the cells. qRT-PCR is not enough to demonstrate viral replication. The authors must show the electrophoresis of full genomic and sub-genomic viral RNAs.

The level of replication is consistently expressed in “fold-change”. What does it mean? Although it is stated that MOI was equal to 1 unless indicated (line 154), what is the viral load in the inoculum since we don’t know the MOI in figures 1A, 2, 4? Most of the time, the authors observed an increase by 2 to 3-fold of the genome copies by PCR. It could very well be artefactual, and I feel that the increase range that has been observed is below the precision level of the technique. Indeed, 2 and 3 fold-increases represent 0.3 and 0.47 log increase, respectively. This is not much and this cannot be accurately measured by qRT-PCR.

How come the replication level was similar for MOI ranging from 1 to 100 (lines 298-299 and figure S2)?

Figures 3A and 3B: ELISA are not quantitative assays and cannot be used to quantify viral protein synthesis. Western blot analysis of the viral products during a time course analysis should be preferred because it an accepted method among virologist for demonstrating viral protein synthesis during the replication. The detection of VP1 structural protein is not good enough, and the detection of a non-structural protein should be added.

Figure 3C: One would not see any remarkable shift between mocked and infected cells during the experiments.

Figure 3D: it is hard to be convinced that there is NoV replication looking at the IFA.

Lines 316-326: it is hard to be convinced by the role of trypsin during NoV replication. Did the authors analyze the role of bile acid? Bile acid has been shown to enhance not only NoV replication in enteroids but also replication of porcine enteric calicivirus.

Minor comments:

Line 277: what is p48?

Line 336: what is a modest role?

Reviewer 2 Report

Human norovirus (HuNoV) is the no.1 cause of gastroenteritis worldwide. Historically, this virus does not grow in cell culture. Recently, human B cells and human intestinal enteroid (HIE) cell culture were shown to support the replication of HuNoV. However, it is still unclear whether these culture systems are robust enough for HuNoV biomedical research. Therefore, exploration of other culture system is still badly needed. In this study, the author showed that Vero cells support the replication of two pandemic GII.4 HuNoV strains and one GII.3 strains. Addition of trypsin can increase HuNoV replication. This is a very exciting new development in the norovirus field although the level of HuNoV replication in Vero cells seems moderate. I fully support the publication of this manuscript. I think this is the first step toward the development of a simple culture system for HuNoV, which can be easily adopted by all the labs in the world. It is highly encouraging. The manuscript is generally well-written and the findings are exciting. I have several minor points which need to be addressed.

(1) The increase in RNA copies was modest (2-5 fold). Please provide detailed information how those data were calculated.

(2) Fig.3C, upper panel should have error bar. Fig.3D, the resolution for this image seems low.

(3) have the authors tried to knock down several ISGs together and see if HuNoV replication further increases.

(4) Can authors detect HuNoV VP1 and nonstructural proteins in HuNoV-infected Vero cells by Western blot?

(5) Please discuss what strategies may further enhance HuNoV replication in Vero cells.